# Deletions of *CDKN2A* and *MTAP* Detected by Copy-Number Variation Array Are Associated with Loss of p16 and MTAP Protein in Pleural Mesothelioma

**DOI:** 10.3390/cancers15204978

**Published:** 2023-10-13

**Authors:** Bart Vrugt, Michaela B. Kirschner, Mayura Meerang, Kathrin Oehl, Ulrich Wagner, Alex Soltermann, Holger Moch, Isabelle Opitz, Peter J. Wild

**Affiliations:** 1Institute of Pathology and Molecular Pathology, University Hospital Zurich, 8091 Zurich, Switzerlandulrich.wagner@usz.ch (U.W.); alex.soltermann@patholaenggasse.ch (A.S.); holger.moch@usz.ch (H.M.); 2Department of Thoracic Surgery, University Hospital Zurich, 8091 Zurich, Switzerland; michaela.kirschner@usz.ch (M.B.K.); mayura.meerang@usz.ch (M.M.); isabelle.schmitt-opitz@usz.ch (I.O.); 3Dr. Senckenberg Institute of Pathology (SIP), University Hospital Frankfurt, Goethe University, 60590 Frankfurt am Main, Germany; peter.wild@kgu.de; 4Wildlab, University Hospital Frankfurt MVZ GmbH, 60596 Frankfurt am Main, Germany; 5Frankfurt Institute for Advanced Studies (FIAS), 60438 Frankfurt am Main, Germany

**Keywords:** pleural mesothelioma, p16, MTAP, molecular pathology, diagnostics, immunohistochemistry, copy-number array, CDKN2A

## Abstract

**Simple Summary:**

Loss of the *CDKN2A* gene and its protein p16 is a common event in pleural mesothelioma, which is frequently investigated to confirm the diagnosis. The loss of *CDKN2A* is often accompanied by simultaneous loss of the genes located in the vicinity, including the *MTAP* gene and protein. Detection of *CDKN2A* loss is usually achieved by relatively expensive fluorescent in situ hybridization (FISH) analysis. Here, we investigated if inexpensive immunohistochemistry (IHC) for p16 and MTAP could be used as an alternative detection method. Comparing *CDKN2A* and *MTAP* gene status, analyzed by the copy-number variation array, with p16/MTAP IHC revealed high sensitivity and specificity of the IHC for detecting gene loss. Our data show that p16/MTAP IHC can be used as an alternative to p16 FISH for detection of *CDKN2A/MTAP* loss. We recommend combined MTAP and p16 immunohistochemistry to confirm the diagnosis of PM.

**Abstract:**

*CDKN2A* deletion is a common alteration in pleural mesothelioma (PM) and frequently associated with co-deletion of *MTAP*. Since the standard detection method for *CDKN2A* deletion and FISH analysis is relatively expensive, we here investigated the suitability of inexpensive p16 and MTAP IHC by comparing concordance between IHC and OncoScan CNV arrays on samples from 52 PM patients. Concordance was determined using Cohen’s kappa statistics. Loss of *CDKN2A* was associated with co-deletion of *MTAP* in 71% of cases. *CDKN2A-MTAP* copy-number normal cases were also IHC positive in 93% of cases for p16 and 100% for MTAP, while homozygous deletion of *CDKN2A-MTAP* was always associated with negative IHC for both proteins. In cases with heterozygous *CDKN2A-MTAP* loss, IHC expression of p16 and MTAP was negative in 100% and 71%, respectively. MTAP and p16 IHC showed high sensitivity (MTAP 86.5%, p16 100%) and specificity (MTAP 100%, p16 93.3%) for the detection of any gene loss. Loss of MTAP expression occurred exclusively in conjunction with loss of p16 labeling. Both p16 and MTAP IHC showed high concordance with Oncoscan CNV arrays (kappa = 0.952, *p* < 0.0001, and kappa = 0.787, *p* < 0.0001 respectively). We recommend combined MTAP and p16 immunohistochemistry to confirm the diagnosis of PM.

## 1. Introduction

Pleural mesothelioma (PM) is an aggressive neoplasm arising from the mesothelial cells lining the pleural cavity and is strongly associated with previous exposure to asbestos [1]. The algorithm for diagnosis of malignant PM requires histomorphologic evidence of invasion and immunohistochemical evidence of the mesothelial nature of the tumor [2]. However, in cases of small biopsy specimens, in situ mesothelial proliferations, and cell blocks, this approach is insufficient to make a clinically meaningful diagnosis. In such cases, additional studies such as fluorescence in situ hybridization (FISH) of cyclin-dependent kinase inhibitor 2A (*CDKN2A*) and BRCA-1-associated protein-1 (BAP1) immunohistochemistry are useful to confirm the malignant nature of the lesion [3,4,5,6,7,8,9,10]. Despite optimal specificity, *CDKN2A* FISH has variable sensitivity in PM with preserved BAP1 expression, ranging from 58% to 80% depending on the study and histological subtype of PM [7,8,9,10]. 

The *CDKN2A* locus encodes the p16 protein and resides on chromosome 9p21 in proximity to a cluster of genes harboring *CDKN2B* and methylthioadenosine phosphorylase (*MTAP*). MTAP is a key enzyme that cleaves methylthioadenosine, resulting in precursor substrates required for adenosine and methionine salvage pathways [11]. *CDKN2A* deletion is a frequent alteration in PM, occurring in up to 74% of cases [12,13,14,15,16,17]. Due to its proximity to *CDKN2A,* homozygous deletion of *MTAP* occurs frequently in various tumors, including mesotheliomas. This is associated with low levels of mRNA of both genes, which may explain the concurrent loss of p16 and MTAP protein expression and has opened the possibility of IHC instead of FISH to determine *CDKN2A* gene status [14]. Disadvantages of *CDKN2A* FISH are technical complexity, longer turnaround times, and higher costs. In contrast to FISH and other molecular tests, IHC is a technically feasible and cost-effective substitute that can be applied in any routine lab. Some early studies showed poor agreement between p16 and MTAP protein expression and *CDKN2A* status determined by FISH, suggesting that p16 and MTAP IHC are not reliable biomarkers for detecting *CDKN2A* loss [6,15]. Recently, however, evidence has accumulated that loss of MTAP protein expression is associated with a homozygous deletion of *CDKN2A* determined by FISH [5,16,17,18,19]. The aim of this study is to evaluate the diagnostic utility of MTAP and p16 immunohistochemistry in the simultaneous detection of *CDKN2A* and *MTAP* alterations using the copy-number variation (CNV) assay.

## 2. Materials and Methods

### 2.1. Tissue Samples and Tissue Microarray Construction

All patients gave written informed consent, the study was approved by the cantonal Ethics Committee Zurich (KEK-ZH-Nr. 2012-0094), and the study was conducted in accordance with the Declaration of Helsinki. Formalin-fixed paraffin-embedded tissue (FFPE) blocks containing diagnostic biopsies of 52 PMs between 1999 and 2015 were retrieved from the archives of our pathology department. Tissue processing was performed overnight using 4% neutral buffered formalin. Corresponding slides of all cases were independently reviewed by two experienced pathologists (B.V. and A.S.) and histologically categorized into epithelioid, biphasic, and sarcomatoid PM according to the latest WHO guidelines [20]. Additional staining, including mesothelial (CK5/6, calretinin) and epithelial markers (Ber-Ep4, Claudin-4), was performed to confirm the mesothelial nature of the tumor. In cases with a biphasic or sarcomatoid PM, additional pancytokeratin and podoplanin (D2-40) stains were applied [2]. 

Tissue blocks were used for array-based genome-wide copy-number analysis and construction of a tissue microarray (TMA). TMA construction was accomplished with a custom-made, semiautomatic tissue arrayer (Beecher Instruments, Sun Prairie, WI, USA) as previously described [21]. Only representative tumor blocks containing sufficient tumor tissue to perform CNV array and construction of TMA were selected, and four tissue cores, 0.6 mm in diameter, were taken from each case and transferred into a recipient paraffin block. In order to capture the diversity of PM, cores were taken from separately marked areas. 

### 2.2. Copy-Number Variation Arrays (Oncoscan)

Array-based genome-wide copy-number analysis was conducted using OncoScan FFPE microarrays (Affymetrix, Santa Clara, CA, USA) as previously described [3]. Considering potential spatial heterogeneity cores were taken from separately marked areas within the tumor. From each patient, samples of normal tissue were collected and analyzed in parallel with tumor samples.

### 2.3. Immunohistochemistry for p16 and MTAP

IHC was performed using an automated single-staining procedure (Benchmark Ultra; Ventana Medical Systems, Oro Valley, AZ, USA). Antigen retrieval for p16 and MTAP was performed using Ventana CC1/EDTA buffer for 48 and 32 min, respectively. Using negative and positive on-slide-controls, including reactive mesothelial proliferations, 4 µm sections of the TMA were stained with monoclonal antibodies against p16 (clone E6H4, ready-to-use, Ventana Medical Systems-Tucson, Oro ValleyAZ, USA; Roche) incubated for 4 min at 37 °C and MTAP (clone EPR6893, Abcam Limited, Hangzhou, China) at 1:500 dilution, incubated for 32 min at room temperature. 

Any p16 nuclear staining with or without cytoplasmatic staining was regarded as positive, while loss of p16 nuclear immunoreactivity was considered negative. Strong nuclear and/or cytoplasmatic staining for MTAP was interpreted as positive, whereas complete loss of cytoplasmic labeling was considered negative. Because of the variable expression of MTAP, staining was regarded as negative when staining intensity was weaker than the internal positive control, such as stromal or inflammatory cells. Heterogenous expression of p16 or MTAP was regarded as positive, and the percentage of positively stained cells was recorded. To validate the immunohistochemical results *CDKN2A* FISH analysis on gross slides was performed in selected cases, two with homozygous, two with heterozygous deletion, and two with retained *CDKN2A* as determined by CNV. Homozygous deletion by FISH was defined as 20% or more of 100 nuclei showing a loss of both *CDKN2A* signals, Loss of one *CDKN2A* signal in more than 50% together with less than 20% loss of both signals was regarded as heterozygous deletion.

### 2.4. Statistical Analysis

In order to determine the suitability of p16 and MTAP IHC for detecting loss of p16 and MTAP protein expression in samples with known homozygous or heterozygous deletion determined by Oncoscan CNV arrays, sensitivity and specificity were calculated. Furthermore, for assessment of the concordance between IHC and CNV, Cohen’s kappa was calculated. All calculations were performed using IBM SPSS Statistics 26. *p* < 0.05 was considered as statistically significant. Receiver operating characteristic (ROC) curves were generated using GraphPad Prism v.9 (GraphPad Software, Boston, MA, USA)

## 3. Results

Histological examination of the diagnostic biopsies revealed 43 epithelioid (82.7%), 6 biphasic (11.5%), and 3 sarcomatoid PM (5.8%). Basic patient demographics are summarized in Table 1, while the results of the IHC and CNV arrays are summarized in Figure 1.

### 3.1. Copy-Number Variation Array (Oncoscan)

CNV analysis identified homozygous deletion of both *CDKN2A* and *MTAP* in 20 cases (38%). In 14 cases (27%), heterozygous deletion of both *CDKN2A* and *MTAP* was found. In three cases (6%), homozygous deletion of *CDKN2A* was accompanied by a heterozygous loss of the *MTAP* gene. Fifteen cases (29%) revealed normal *CDKN2A* and *MTAP* copy numbers. Copy-number loss of *MTAP* in the absence of *CDKN2A* deletion was not detected. 

Heterozygous loss of both *CDKN2A* and *MTAP* was identified in 15 and 14 cases of epithelioid PM. In one case of epithelioid PM, a homozygous loss of *CDKN2A* combined with a heterozygous loss of *MTAP* was detected. Fifteen cases with epithelioid PM revealed normal copy numbers of both genes were found. In five biphasic and two sarcomatoid PM, homozygous deletion of *CDKN2A* and *MTAP* was noted. One biphasic and one sarcomatoid PM showed homozygous *CDKN2A* loss combined with a heterozygous deletion of *MTAP*. None of the cases with a biphasic or sarcomatoid PM showed a WT pattern.

### 3.2. Concordance of Immunohistochemistry and Copy-Number Variation Array

Consistent with the immunoreactivity in the external controls, p16 positivity in the PM cases was characterized by a diffuse nuclear and cytoplasmatic staining pattern. Strong expression of MTAP was observed in 20 cases. In comparison to stromal cells, MTAP-negative cases showed a faint to moderately cytoplasmic staining (example shown in (Figure 2B)). 

PM with a wild-type (WT) *CDKN2A-MTAP* showed retained strong expression of MTAP in all cases, whereas p16 positivity was observed in all but one WT case. None of the WT cases showed negativity of both p16 and MTAP (Figure 2A). Conversely, homozygous loss of both *CDKN2A* and *MTAP* was associated with loss of p16 and MTAP expression in all cases (Figure 2B). Heterozygous loss of *CDKN2A* and *MTAP* was observed in 14 cases, which was accompanied by negative staining of p16 and MTAP in 100% and 71%, respectively (Figure 2C). In three cases, homozygous loss of *CDKN2A* and heterozygous loss of *MTAP* were associated with loss of p16 in all and MTAP expression in two cases. Similar to *CDKN2A-MTAP* by CNV, retained expression of p16 in the absence of MTAP was not observed in any of the cases. *CDKN2A* FISH in the six selected cases corresponded with the results of the CNV array. 

Nine mesotheliomas (17%) showed a heterogeneous p16 and MTAP expression, with labeling of both markers in a proportion of cores or tumor cells ranging from 10 to 90% of cells. Six of these cases were WT, as determined by the CNV array. Gross slides of these cases showed spatial loss of both p16 and MTAP (Figure 3), ranging from a focal loss in a smaller cell cluster (Figure 3B) to up to 50% loss in some tumor areas (Figure 3A). In one case with a homozygous deletion of both *CDKN2A* and *MTAP* and one case with a homozygous *CDKN2A* deletion and heterozygous *MTAP* deletion, strong MTAP expression was seen in 10% of the tumor cells, whereas p16 was negative in both cases. In one mesothelioma with heterozygous *CDKN2A-MTAP* deletion, positive MTAP expression was seen in 60% of the tumor cells, whereas p16 was completely negative.

### 3.3. Sensitivity and Specificity of p16 and MTAP Immunohistochemistry

P16 and MTAP immunohistochemistry showed a specificity of 100% and 93.3% for detection of *CDKN2A* and *MTAP* copy-number alterations detected by CNV array, respectively. Compared to the results of the CNV array, sensitivity of p16 (100%) was better than that of MTAP (86.5%) in detecting any type of loss (homozygous or heterozygous). Concordance (Cohen’s kappa) for detection of any loss between Oncoscan CNV array and IHC was 0.952 for p16 and 0.787 for MTAP (*p* < 0.0001 for both). P16 IHC showed the same sensitivity for detection of both homozygous and heterozygous loss as determined by CNV. In the case of MTAP, however, sensitivity for detection of a homozygous loss was 100%, while a heterozygous loss was only detected by IHC with a sensitivity of 70.6%. Accordingly, Cohen’s kappa for detection of a heterozygous *MTAP* loss was reduced to 0.692 (*p* > 0.0001), while kappa for identifying homozygous *MTAP* deletion reached a perfect value of 1.0 (*p* > 0.0001). Sensitivity, specificity, and Cohen’s kappa are summarized in Table 2. ROC curves and the corresponding AUCs are depicted in Figure 4.

## 4. Discussion

Using a CNV array for genome-wide detection of copy-number alterations, we have shown for the first time that loss of *CDKN2A* in PM is consistently associated with co-deletion of the *MTAP* gene. The genomic loss was also associated with loss of both p16 and MTAP protein expression in all cases with homozygous deletion of *CDKN2A-MTAP* and in 100% and 71% with heterozygous deletion of *CDKN2A* and *MTAP*, respectively. Compared with CNV results, p16 IHC showed a sensitivity of 100% and a specificity of 93.3%, while MTAP IHC showed a sensitivity of 86.5% and a specificity of 100% for detecting loss. These data support the concept that combined p16 and MTAP immunohistochemistry parallels *CDKN2A*-*MTAP* gene status. 

In this study, *CDKN2A*-*MTAP* co-deletion was associated with loss of both p16 and MTAP protein expression in all cases with homozygous deletion and in the majority of cases with heterozygous deletion of *CDKN2A-MTAP*. Consistent with our findings, various studies using technologies, including FISH, NGS, and single-nucleotide polymorphism microarray, have shown that homozygous *CDKN2A* deletion is associated with *MTAP* co-deletion and that isolated *MTAP* deletion does not occur without concurrent *CDKN2A* loss [12,13,22]. Using FISH, Illei et al. [12] observed discordant deletion of *CDKN2A* without *MTAP* loss in 9% of their cases, a finding that possibly reflects the heterozygous *MTAP* deletion combined with homozygous *CDKN2A* loss found in three of our cases (6%). This supports the concept that loss of MTAP deficiency is only present in the context of a homozygous loss of *CDKN2A*, specifically when the 9p21.3 deletion is large enough to encompass the *MTAP* gene. 

Homozygous deletion of *CDKN2A* determined by FISH is formally considered to be diagnostic of malignancy if more than 20% of the tumor nuclei show loss of both 9p21 signals [6]. However, FISH results are highly variable ranging from no detectable loss and heterozygous deletion to homozygous deletion of *CDKN2A* both in malignant as well in reactive mesothelial proliferations, which is mainly attributable to artifactual loss of signals because of nuclear sectioning [23,24,25]. Previous published work investigating p16 protein expression as a substitute for *CDKN2A* gene status has shown a poor concordance between p16 immunohistochemistry and homozygous *CDKN2A* deletion by FISH in PM indicating that p16 labeling cannot serve as a surrogate for *CDKN2A* genomic loss [6]. In contrast to these results, but consistent with recent observations in peritoneal mesotheliomas [13], we found loss of p16 expression in all cases with homozygous *CDKN2A* deletion determined by CNV analysis. Similar to FISH, we cannot rule out that loss of p16 protein in the 14 cases with heterozygous deletion of *CDKN2A* is the result of spatial intra-tumoral heterogeneity [24]. Other mechanisms, such as mini deletions not detected by the CNV array, point mutations, or DNA methylation of the p16 promoter gene may also be attributable to loss of p16 labeling [26,27]. 

The poor concordance between p16 expression and *CDKN2A* loss has shifted attention to MTAP IHC as a surrogate marker for *CDKN2A* gene status. In an initial study, Zimling et al. [15] observed loss of MTAP expression in 65% and 23% of the malignant and benign mesothelial proliferations, suggesting that MTAP IHC is not useful for PM diagnostics. However, in recent years, there has been a growing body of evidence for high specificity of MTAP protein expression for detection of homozygous *CDKN2A* deletion by FISH [5,16,17,18,19]. MTAP turned out to be 100% specific for diagnosing PM, with sensitivity ranging from 43 to 65%. Consistent with these observations, MTAP immunohistochemistry in our study was 86.5% sensitive and 100% specific for *CDKN2A-MTAP* gene status. As mentioned before, *MTAP* deletion without *CDKN2A* co-deletion did not occur in any of the PM tumor samples, and loss of MTAP expression occurred exclusively in conjunction with loss of p16 labeling. Positive staining for p16 thus excludes loss of MTAP. Because the interpretation of MTAP immunohistochemistry can be challenging [18], the addition of p16 staining helps to better distinguish MTAP-positive from -negative cases. According to the literature, MTAP deficiency is due to either deletion or epigenetic silencing of the *MTAP* gene [28,29]. Therefore, tests for detecting genomic deletion, including FISH and CNV, may underestimate *MTAP* loss in PM. Therefore, the most appropriate test for demonstrating MTAP deficiency would be detection of loss of protein expression by immunohistochemistry. In our cases, different patterns of combined *CDKN2A*-*MTAP* deletion were associated with complete loss of p16 IHC independent of whether homozygous or heterozygous loss was present. In the case of MTAP, however, loss of protein expression was observed in 100% of cases with homozygous loss, but also in 71% (12/17) cases with a heterozygous *MTAP* deletion. Accordingly, the concordance between IHC and CNV only reached a kappa of 0.692 for detection of a heterozygous loss. Of the 17 heterozygous *MTAP* cases, 3 were associated with a homozygous deletion and 14 with a single-copy loss of *CDKN2A*. Recently, Chapel et al. [22] presented similar data with loss of MTAP expression in 10 of 15 (67%) tumors with heterozygous *MTAP* deletion. Based on the heterogenous MTAP staining in a subset of cases with *MTAP* deletion, they hypothesized the presence of topographical subclones with *MTAP* deletion. Another explanation for the loss of MTAP expression in those cases with heterozygous *MTAP* deletion could be *MTAP* silencing, as has been described in melanoma, where epigenetic dysregulation of *MTAP* and other genes contribute to tumor progression and invasion. We did not find any reports on *MTAP* mutations or mini deletions, which could explain loss of MTAP protein expression. However, in knockout mice heterozygous for MTAP appear to be indistinguishable from WT mice, but died because of T-cell lymphoma. In comparison with normal controls, levels of MTAP, RNA, and MTAP expression by IHC were significantly reduced in the tissues in these mice infiltrated with lymphoma [29]. Whether these observations can be extended to heterozygous *MTAP* deletions in PM remains to be elucidated.

In the present study, topographical loss of both p16 and MTAP protein in part of the tumor tissue was observed in 17% of our cases, the majority of which were WT (66%) according to CNV analysis (Figure 2). Also, in gross slides of WT cases, small groups of p16-MTAP-negative tumor cells could be identified between aggregates expressing both p16 and MTAP. Consistent with our findings heterogeneity of MTAP staining intensity in PM has been reported in several studies [5,18,22]. Berg and co-workers [5] described two cases with negative MTAP expression in 50% and 75% of the tumor cells (similar to the case described in Figure 2), whereas Chapel et al. [18] found a heterogeneous MTAP retention in four cases with homozygous *CDKN2A* deletion determined by FISH. Using combined p16 and MTAP IHC, we were able to discriminate tumor areas with sustained p16-MTAP expression from tumor cell aggregates that were negative for both markers. In three cases (6%), a heterozygous *MTAP* deletion combined with homozygous *CDKN2A* deletion was found, which was associated with loss of both p16 and MTAP expression in two cases. The concurrent biallelic loss of *CDKN2A* next to heterozygous *MTAP* gene deletion probably reflects genetic tumor progression and characterizes the spatial heterogeneity of PM [24]. 

This study has several limitations. First, it is difficult to compare FISH and CNV in assessing *CDKN2A* and *MTAP* copy-number status in PM. The FISH assay is still regarded as the gold standard for detection of copy-number loss in PM, because it allows visual identification of homozygous and heterozygous deletions. Nevertheless, molecular assays, such as CNV, enable the simultaneous analysis of both *CDKN2A* and *MTAP* of more cells, including the distinction of single- and two-copy-number loss, and are not compromised by artificial loss of signals due to cutting of sections. Second, The number of female patients was relatively low (16%) but comparable with those found in the general population (17%) [20]. In a larger cohort of 222 patients, who had been treated in our hospital, 23 (10%) were females. Loss of MTAP protein was found in 47% of the female cases, which is comparable with the findings in the present cohort (43%). Third, several studies have used a three-tiered system to assess the H-score of MTAP expression or, for statistical reasons, applied different cut-offs for positive or negative staining [15,16]. In this study, labeling in a proportion of tumor cells in the TMA was interpreted as positive. Although in the vast majority of cases p16 and MTAP were either positive or negative, the diagnostic significance of heterogeneous p16 and MTAP staining in small biopsies or cytological specimens remains unclear. In such cases, the detection of copy-number loss by p16 FISH, especially of those tumor aggregates showing loss of both p16 and MTAP, may help to determine the malignant nature of the mesothelial proliferation.

## 5. Conclusions

In conclusion, combined p16 and MTAP immunohistochemistry is a cost-effective surrogate for *CDKN2A-MTAP* genetic alterations that can be reliably used to detect homozygous deletions and visualize the spatial heterogeneity of the *CDKN2A-MTAP* gene complex within the same tumor sample. Topographic labeling of both MTAP and p16 in the same tumor areas of cases with a heterogeneous staining pattern indicates a common mechanism of inactivation of *CDKN2A* and *MTAP* genes. To obtain a reliable diagnosis of PM, combined MTAP and p16 immunohistochemistry is recommended, especially in BAP-1-positive cases, small biopsies, and cell blocks.

## Figures and Tables

**Figure 1 cancers-15-04978-f001:**
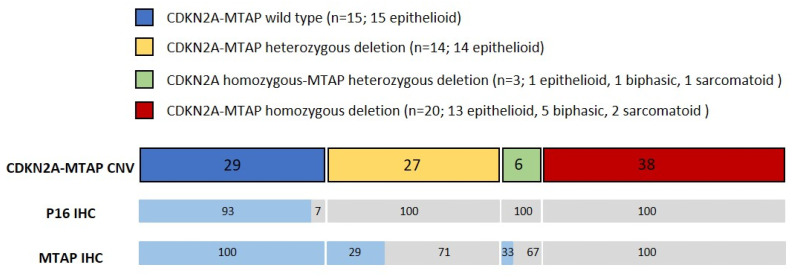
Comparison of p16/MTAP immunohistochemistry (IHC) and *CDKN2A-MTAP* copy-number loss determined by CNV array. Numbers indicate percentage of each group compared to total number of patients (n = 52) for CNV, and to number of patients from each group for IHC, respectively. Blue represents IHC positive, and gray represents IHC negative cases.

**Figure 2 cancers-15-04978-f002:**
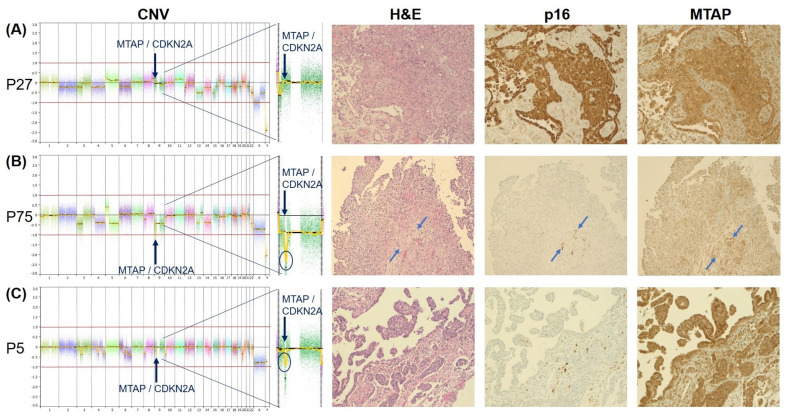
CNV profiles and IHC staining patterns of p16 and MTAP in a case with WT ((**A**), P27), a case with homozygous *CDKN2A-MTAP* deletion ((**B**), P75), and a case with heterozygous *CDKN2A-MTAP* deletion ((**C**), P5), respectively. While the case with homozygous loss (**B**) shows consistent loss of MTAP staining, MTAP immunoreactivity is retained in some cases with heterozygous loss, as shown here in (**C**). Note the positive internal control for p16 and MTAP of stromal cells between the negative tumor cell infiltrates as marked by blue arrows in (**B**). CNV results are shown with a zoom-in into chromosome 9, which harbors the *CDKN2A* and *MTAP* loci on q21.3. H&E, p16, and MTAP staining are shown in 20× magnification. (Appendix A provides additional information).

**Figure 3 cancers-15-04978-f003:**
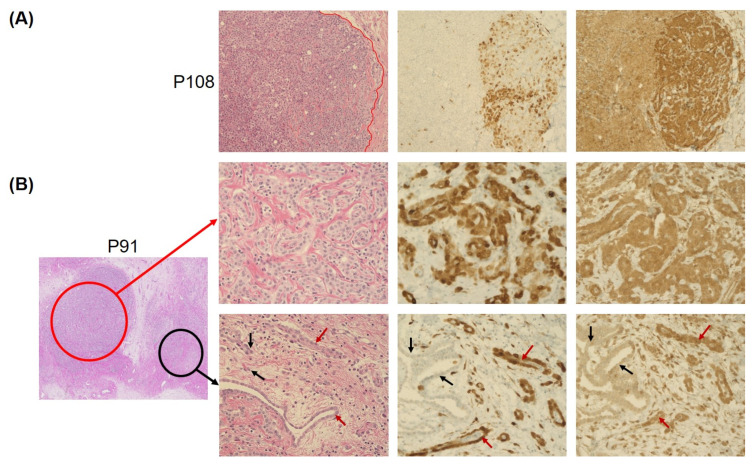
Heterogeneity of p16 and MTAP protein expression in PM observed in 17% of cases. (**A**) shows an example of a tumor sample with wild-type *CDKN2A/MTAP* and retained protein expression with very high spatial heterogeneity, and p16 and MTAP negativity in up to 50% of tumor cells (tumor area left of red line). The wild-type case presented in (**B**), however, showed the expected 100% p16 and MTAP positivity in one area of the obtained section (red circle, middle row), combined with only focal loss of both p16 and MTAP in around 10% of the tumor cells in a different area (black circle, bottom row). Staining positive tumor cells are marked with red arrow, while staining negative tumor cells are marked with black arrows. (Magnification: 20×).

**Figure 4 cancers-15-04978-f004:**
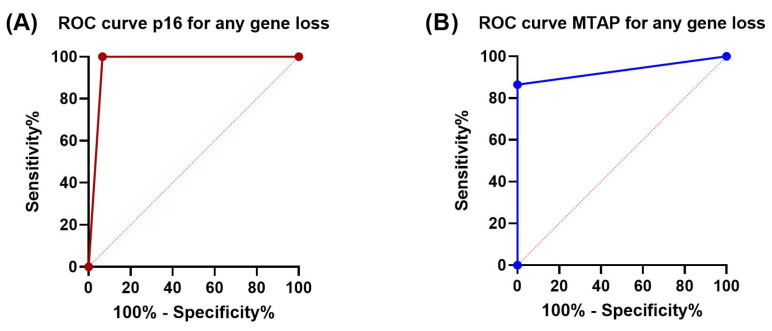
ROC curves showing diagnostic sensitivity and specificity of p16 and MTAP IHC for homozygous and heterozygous loss of *CDKN2A* and *MTAP* genes: (**A**) P16 sensitivity 100%, specificity 93%, AUC 0.97 (95% CI: 0.89–1.0); (**B**) MTAP sensitivity 86.5%, specificity 100%, AUC 0.93 (95% CI: 0.86–1.0). AUC is annotaed with 95% CI using the Wilson/Brown method.

**Table 1 cancers-15-04978-t001:** Basic patient demographics.

Median Age (Range)	60 (35–73)
Gender	
Male	45 (86.5%)
Female	7 (13.5%)
Histotype	
Epithelioid	43 (82.7%)
Biphasic	6 (11.5%)
Sarcomatoid	3 (5.8%)
Pathological Stage (IMIG, 8th edition)	
IA	2 (3.8%)
IB	23 (44.2%)
II	0 (0.0%)
IIIA	8 (15.4%)
IIIB	15 (28.8%)
unknown	4 (7.7%)
Median OS from diagnosis (range)	14.95 months (3.94–79.86)

**Table 2 cancers-15-04978-t002:** Overview of concordance, sensitivity, and specificity of IHC.

	Any Loss		Homozygous Loss		Heterozygous Loss	
	p16	MTAP	p16	MTAP	p16	MTAP
Cohen’s kappa	0.952	0.787	0.944	1.0	0.931	0.692
Sensitivity	100%	86.5%	100%	100%	100%	70.6%
Specificity	93.3%	100%	93.3%	100%	93.3%	100%

## Data Availability

The data presented in this study are available on request from the corresponding author.

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
