# Peer review of "Deletions of CDKN2A and MTAP Detected by Copy-Number Variation Array Are Associated with Loss of p16 and MTAP Protein in Pleural Mesothelioma"

_cancers, 2023, doi:10.3390/cancers15204978_

Round 1
Reviewer 1 Report
The authors Vrugt et al have developed a well constructed manuscript recommending the use of combined MTAP and p16 immunohistochemistry to confirm the diagnosis of pleural mesothelioma.
Their study clearly examines and reports on the concordance of IHC and CNV for p16 and MTAP that is vital to improving current knowledge on the diagnosis of pleural mesothelioma. In particular, their discussion and addressing the issue of spatial heterogeneity on pleural mesothelioma are commendable. I have only 2 issues that require changes.
1. On line 261, the authors have referenced an article by H H Wu in the AJCP. I believe the correct paper to reference is by Di Wu in AJCP 2013 139(1) 39-46
2. There is a typo on line 353 "fast"
Overall an important conclusion as IHC is far more cost effective than other molecular tests used in clinical practice currently.
I would recommend only minor changes as mentioned in my previous comment on the language quality of the manuscript.
Reviewer 2 Report
Authors investigated whether p16 and MTAP IHC could be used as an alternative detection method of FISH to confirm the diagnosis of mesothelioma. They showed high sensitivity and specificity for detecting gene loss by IHC. The manuscript was well-written and easy to follow.
Several points to consider:
1. Line 111: Detailed IHC staining should be included for reproducing the findings by others (e.g., antibody dilution and incubation time, and reagent used for antigen retrieval). The age of FFPE blocks and tissue fixation conditions should be mentioned since these factors will affect the IHC staining results, which may lead to the inconsistency of the findings from different labs.
2. Line 322: Have the authored considered other possibilities for the loss of MTAP expression in those cases with heterozygous MTAP deletion (e.g., MTAP mutation)? Small deletion or mutations of MTAP cannot be detected by CNV. It would be helpful if authors can confirm the CNV data using FISH for 14 cases with heterozygous MTAP deletion.
3. Line 353: “Although in the fast majority of cases…….”. Should it be “….vast majority of cases…..”?
4. The number of female patients was much lower than males (7 vs. 45) in the analysis. It would be good to follow up with more female patients to confirm the findings.
5. Recommend authors to perform ROC analysis for specificity and sensitivity assessment of IHC testing performance.
Reviewer 3 Report
Comments to the authors
The manuscript by Vrugt B et al. entitled “Deletions of CDKN2A and MTAP detected by CNV array are associated with loss of p16 and MTAP protein in pleural mesothelioma” showed that the combined use of MTAP and p16 IHC could be more beneficial for the accurate diagnosis of PM.
Although the conclusion of this study is not novel, the experimental data are generally convincing. The authors need to clarify some issues before the publication in the journal of Cancers.
Major comments;
1. The authors should provide the details of the way of IHC because the IHC staining is sometimes influenced by the methods including antigen retrieval.
2. Regarding IHC, the authors had better show the image of negative and positive control slide.
3. According to the data shown in Table 2, p16 IHC seems to be enough to detect heterozygous or homozygous loss of CDKN2A. Please provide the merit of combined use of MTAP staining in addition to p16 staining.
Minor comments;
1. The discussion section is a little too long. It would be better to avoid redundancy and shorten the context.
none
Round 2
Reviewer 3 Report
The authors have clarified my concerns that were raised at the initial submission and revised well.
None